META-RESEARCH

# Why we need to report more than 'Data were Analyzed by t-tests or ANOVA'

**Abstract** Transparent reporting is essential for the critical evaluation of studies. However, the reporting of statistical methods for studies in the biomedical sciences is often limited. This systematic review examines the quality of reporting for two statistical tests, t-tests and ANOVA, for papers published in a selection of physiology journals in June 2017. Of the 328 original research articles examined, 277 (84.5%) included an ANOVA or t-test or both. However, papers in our sample were routinely missing essential information about both types of tests: 213 papers (95% of the papers that used ANOVA) did not contain the information needed to determine what type of ANOVA was performed, and 26.7% of papers did not specify what post-hoc test was performed. Most papers also omitted the information needed to verify ANOVA results. Essential information about t-tests was also missing in many papers. We conclude by discussing measures that could be taken to improve the quality of reporting.
DOI: https://doi.org/10.7554/eLife.36163.001

**TRACEY L WEISSGERBER\*, OSCAR GARCIA-VALENCIA, VESNA D GAROVIC, NATASA M MILIC[†] AND STACEY J WINHAM[†]**

**\*For correspondence:**
weissgerber.tracey@mayo.edu

[†]These authors contributed equally to this work

**Competing interests:** The authors declare that no competing interests exist.

## Introduction

The inability to reproduce key scientific results in certain areas of research is a growing concern among scientists, funding agencies, journals and the public (*Nature, 2013*; *Fosang and Colbran, 2015*; *National Institutes of Health, 2015a*; *National Institutes of Health, 2015b*; *Nature, 2017*). Problems with the statistical analyses used in published studies, along with inadequate reporting of the experimental and statistical techniques employed in the studies, are likely to have contributed to these concerns. Older studies suggest that statistical errors, such as failing to specify what test was used or using incorrect or suboptimal statistical tests, are common (*Müllner et al., 2002*; *Ruxton, 2006*; *Strasak et al., 2007*), and more recent studies suggest that these problems persist. A study published in 2011 found that half of the neuroscience articles published in five top journals used inappropriate statistical techniques to compare the magnitude of two experimental effects (*Nieuwenhuis et al., 2011*). A more recent study of papers reporting the results of experiments that examined the effects of prenatal interventions on offspring found that the statistical analyses in 46% of the papers were invalid because authors failed to account for non-independent observations (i.e., animals from the same litter; *Lazic et al., 2018*). Many studies omit essential details when describing experimental design or statistical methods (*Real et al., 2016*; *Lazic et al., 2018*). Errors in reported p-values are also common and can sometimes alter the conclusions of a study (*Nuijten et al., 2016*).

A main principle of the SAMPL guidelines for reporting statistical analyses and methods in the published literature is that authors should "describe statistical methods with enough detail to enable a knowledgeable reader with access to the original data to verify the reported results" (*Lang and Altman, 2013*). However, these guidelines have not been widely adopted.

Clear statistical reporting also allows errors to be identified and corrected prior to publication. The journal *Science* has attempted to improve statistical reporting by adding a Statistical Board of Reviewing Editors (*McNutt, 2014*). Other

journals, including *Nature* and affiliated journals (*Nature, 2013*; *Nature, 2017*), *eLife* (*Teare, 2016*) and *The EMBO Journal* (*EMBO Press, 2017*) have recently implemented policies to encourage transparent statistical reporting. These policies may include specifying which test was used for each analysis, reporting test statistics and exact p-values, and using dot plots, box plots or other figures that show the distribution of continuous data.

T-tests and analysis of variance (ANOVA) are the statistical bread-and-butter of basic biomedical science research (*Strasak et al., 2007*). However, statistical methods in these papers are often limited to vague statements such as: "Data were analyzed by t-tests or ANOVA, as appropriate, and statistical significance was defined as p<0.05." There are several problems with such descriptions. First, there are many different types of t-tests and ANOVAs. Vague statistical methods deprive reviewers, editors and readers of the opportunity to confirm that an appropriate type of t-test or ANOVA was used and that the results support the conclusions in the paper. For example, if authors use an unpaired t-test when a paired t-test is needed, the failure to account for repeated measurements on the same subject will lead to an incorrect p-value. Analyses that use inappropriate tests give potentially misleading results because the tests make incorrect assumptions about the study design or data and often test the wrong hypothesis. Without the original data, it is difficult to determine how the test results would have been different had an appropriate test been used. Clear reporting allows readers to confirm that an appropriate test was used and makes it easier to identify and fix potential errors prior to publication.

The second problem is that stating that tests were used "as appropriate" relies on the assumption that others received similar statistical training and would make the same decisions. This is problematic because it is possible to complete a PhD without being trained in statistics: only 67.5% of the top NIH-funded physiology departments in the United States required statistics training for some or all PhD programs that the department participated in (*Weissgerber et al., 2016a*). When training is offered, course content can vary widely among fields, institutions and departments as there are no accepted standards for the topics that should

be covered or the level of proficiency required. Moreover, courses are rarely designed to meet the needs of basic scientists who work with small sample size datasets (*Vaux, 2012*; *Weissgerber et al., 2016a*). Finally, these vague statements fail to explain why t-tests and ANOVA were selected, as opposed to other techniques that can be useful for small sample size datasets.

This systematic review focuses on the quality of reporting for ANOVA and t-tests, which are two of the most common statistical tests performed in basic biomedical science papers. Our objectives were to determine whether articles provided sufficient information to determine which type of ANOVA or t-test was performed and to verify the test result. We also assessed the prevalence of two common problems: i) using a one-way ANOVA when the study groups could be divided into two or more factors, and ii) not specifying that the analysis included repeated measures or within-subjects factors when ANOVA was performed on non-independent or longitudinal data.

To obtain our sample two reviewers independently examined all original research articles published in June 2017 (n = 328, *Figure 1*) in the top 25% of physiology journals, as determined by 2016 journal impact factor (see Methods for full details). Disagreements were resolved by consensus. 84.5% of the articles (277/328) included either a t-test or an ANOVA, and 38.7% of articles (127/328) included both. ANOVA (n = 225, 68.6%) was more common than t-tests (n = 179, 54.5%). Among papers that reported the number of factors for at least one ANOVA, most were using a maximum of one (n = 112, 49.8%) or two (n = 69, 30.7%) factors. ANOVAs with three or more factors were uncommon (n = 6, 2.7%). This approach involved a number of limitations. All the journals in our sample were indexed in PubMed and only published English language articles, so our results may not be generalizable to brief reports, journals with lower impact factors, journals that publish articles in other languages, or journals that are not indexed in PubMed. Further research is also needed to determine if statistical reporting practices in other fields are similar to what we found in physiology.

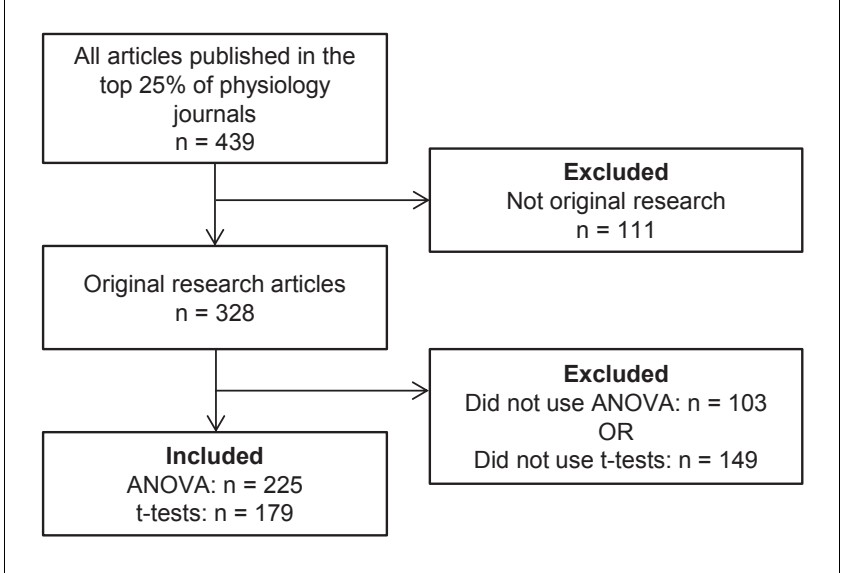

**Figure 1.** Systematic review flow chart.  The flow chart illustrates the selection of articles for inclusion in this analysis at each stage of the screening process.

DOI: https://doi.org/10.7554/eLife.36163.002

The following source data is available for figure 1:

**Source data 1.** Data from systematic review.

DOI: https://doi.org/10.7554/eLife.36163.003

## Can we determine which type of ANOVA was performed?

While ANOVA encompasses a wide variety of techniques, basic biomedical science papers generally use one and two-way ANOVAs, with or without repeated measures. We focused on reporting requirements for these basic tests, as more complex types of ANOVA were rare in our dataset (<3%; *Figure 1—source data 1*). Authors need to report three key pieces of information to allow readers to confirm that the type of ANOVA that was used is appropriate for the study design (see *Box 1* for terminology and *Box 2* for a detailed description of what should be reported). Many papers were missing some or all of this information (*Figure 2*). 213 papers (95% of the papers that used ANOVA) did not contain all information needed to determine what type of ANOVA was performed, including the names and number of factors and whether each factor was entered as a between-subjects or within-subjects factor.

### The number of factors included in the ANOVA and the names and levels of each factor

16.9% of papers (38/225) failed to specify how many factors were included for any ANOVA

performed in the paper. 54% of papers did not specify which factors were included in any ANOVA, whereas fewer than half of papers specified which factors were included for some (n = 13, 5.8%) or all (n = 90, 40%) ANOVAs reported in the manuscript.

Our data suggest that reporting the number, names and levels of factors may be particularly important to determine whether the statistical test is appropriate for the study design. Among papers that used one-way ANOVAs, 60.9% (67/110) used a one-way ANOVA for an analysis where the study design included two or more factors. (Note: Two papers that reported using a maximum of one factor in the ANOVA were excluded, as these papers also included a repeated measures ANOVA with an unknown number of factors.) For example, investigators might have used a one-way ANOVA to compare four independent groups: wild-type mice that received vehicle (placebo), wild-type mice that received a treatment, knockout mice that received vehicle (placebo), and knockout mice that received a treatment. This approach may be appropriate for extremely small sample sizes, as these datasets may not have enough power for an ANOVA with two or more factors. In most cases, however, a two-way ANOVA with mouse strain (wild-type vs. knockout) and treatment (placebo vs. treatment) as factors may provide more information. *Figure 3* and *Figure 4* show the difference between these two approaches and illustrate how the two-way ANOVA may offer additional insight. The two-way ANOVA allows investigators to examine the effects of both factors simultaneously and may help investigators to avoid unnecessary post-hoc tests. If the sample size is large enough, the two-way ANOVA can test for an interaction between the two factors (i.e., whether the effect of mouse strain depends on treatment).

### Whether each factor was entered as a between (independent) or within (non-independent) subjects factor

Among the 18.2% of papers (41/225) that reported using repeated measures ANOVA, 63.4% (n = 26) did not specify whether each factor was entered as a between or within-subjects factor. Many of the 15 papers that provided adequate information reported using one-way repeated measures ANOVA. When the ANOVA only includes one factor, stating that repeated measures were used demonstrates that this factor was treated as a within-subjects or non-independent factor. When a repeated measures

## Box 1. ANOVA terminology

\# ANOVA examines the effect of one or more categorical independent variables, known as 'factors', on a dependent variable. Examples of factors might include age (young vs. old), hypertension (hypertensive vs. normotensive) or time since drug administration (baseline, 1 hour, 6 hours, 12 hours, and 24 hours).

\# The groups or conditions within each factor are called levels. In the example above, age has two levels (young vs. old), hypertension has two levels, and time since drug administration has five levels.

\# Each factor can be entered into the ANOVA as a between-subjects factor or a within-subjects factor. Measurements on unrelated participants, specimens or samples (i.e., age, hypertension) are entered into the ANOVA as between-subjects factors. Longitudinal measurements (i.e., time since drug administration) are entered into the analysis as within-subjects factors.

\# Other types of non-independent measurements may also be entered as within-subjects factors. This includes related measurements that are performed in the same participants (i.e., arm vs. leg), or groups in which subjects are matched (i.e., normotensive and hypertensive participants matched for sex and age).

\# An ANOVA that includes at least one within-subjects factor is called repeated measures ANOVA.

\# A repeated-measures ANOVA with two or more factors may include between-subjects factors, depending on the study design. For example, a two-way repeated measures ANOVA with one between and one within-subjects factor might be used for a study in which men and women completed four weeks of exercise training. Sex (female vs. male) would be a between-subjects factor, whereas exercise training (baseline vs. post-training) would be a within-subjects factor. A two-way repeated measures ANOVA with two within-subjects factors might be used for a study in which vascular function was measured in both arms (right vs. left), before and after 4 weeks of right arm handgrip training (baseline vs. post-training).

\# Post-hoc tests are used to determine which groups in the ANOVA differ from each other. These tests are used when p-values for the ANOVA show that there is a significant effect of any factor, or a significant interaction between two or more factors. There are many different types of post-hoc tests. These tests use different procedures to determine which groups to compare and adjust for multiple comparisons; some are more conservative than others.

DOI: https://doi.org/10.7554/eLife.36163.004

ANOVA includes two or more factors, however, authors need to clearly report which factors were treated as between vs. within-subjects factors. A two-way repeated measures ANOVA could refer to two different tests – an ANOVA with two within-subjects factors, or an ANOVA with one within-subjects factor and one between-subjects factor (*Box 1*).

The remaining 81.8% of papers that included ANOVA did not indicate whether repeated measures were used. Only one of these papers stated that all factors were entered as between-subjects factors (1/184, 0.5%). If repeated measures or within-subjects factors are not mentioned, one would typically assume that all factors were independent or between-subjects. Our data suggest that scientists should be cautious about this assumption, as 28.3% of papers that did not indicate that repeated measures were used (52/184) included at least one ANOVA that appeared to require repeated measures. There is no way to distinguish between papers that used the wrong test and papers that failed to clearly specify which test was used. *Figure 5* illustrates the differences between these two tests. Failing to account for repeated measures in an ANOVA makes it less likely that investigators will detect an effect, as

## Box 2. Checklist for clear statistical reporting for t-tests and ANOVAs

Basic biomedical science studies often include several small experiments. Providing information about statistical tests in the legend of each table or figure makes it easy for readers to determine what test was performed for each set of data and confirm that the test is appropriate.

**t-tests**

# State whether the test was unpaired (for comparing independent groups) or paired (for non-independent data, including repeated measurements on the same individual or matched participants, specimens or samples).

# State whether the test assumed equal or unequal variance between groups.

# Report the t-statistic, degrees of freedom and exact p-value.

# To focus on the magnitude of the difference, it is strongly recommended to report effect sizes with confidence intervals.

**ANOVAs**

# Specify the number of factors included in the ANOVA (i.e., one- vs. two-way ANOVA).

# For each factor, specify the name and level of the factor and state whether the factor was entered as a within-subjects (i.e., independent) factor or as a between-subjects (i.e., non-independent) factor.

# If the ANOVA has two or more factors, specify whether the interaction term was included. If the ANOVA has three or more factors and includes interaction terms, specify which interaction terms were included.

# Report the F-statistic, degrees of freedom and exact p-value for each factor or interaction.

# Specify if a post-hoc test was performed. If post-hoc tests were performed, specify the type of post-hoc test and, if applicable, the test statistic and p-value.

# To focus on the magnitude of the difference, it is strongly recommended to report effect sizes with confidence intervals.

DOI: https://doi.org/10.7554/eLife.36163.005

the variability that can be attributed to individual subjects remains unexplained. To avoid confusion, we recommend that reviewers, editors and journal guidelines encourage authors to specify whether each factor was entered as between or within-subjects factor, even if the paper does not include a repeated measures ANOVA.

*The type of post-hoc test that was performed*

Clear reporting of post-hoc testing procedures allows readers to determine how the authors decided whether to use post-hoc tests, what type of post-hoc tests were performed and whether the tests results were adjusted for multiple comparisons. 72.8% of papers (164/225) stated what test was used to examine pairwise differences after performing an ANOVA (such as

Tukey and Bonferroni). The remaining 27.1% of papers (61/225) did not specify what type of post-hoc tests were performed.

## Can we determine what type of t-test was performed?

Many of the 179 papers that included a t-test were missing information that was needed to determine what type of t-test was performed, including whether the tests were paired or unpaired, or assumed equal or unequal variance.

*Unpaired vs. paired t-tests*

Over half of the papers (53%; 95/179) did not specify whether any t-tests were paired or unpaired; a small proportion (2.2%; 4/179) provided this information for some but not all of the t-tests in the paper. 69 of the 95 papers without

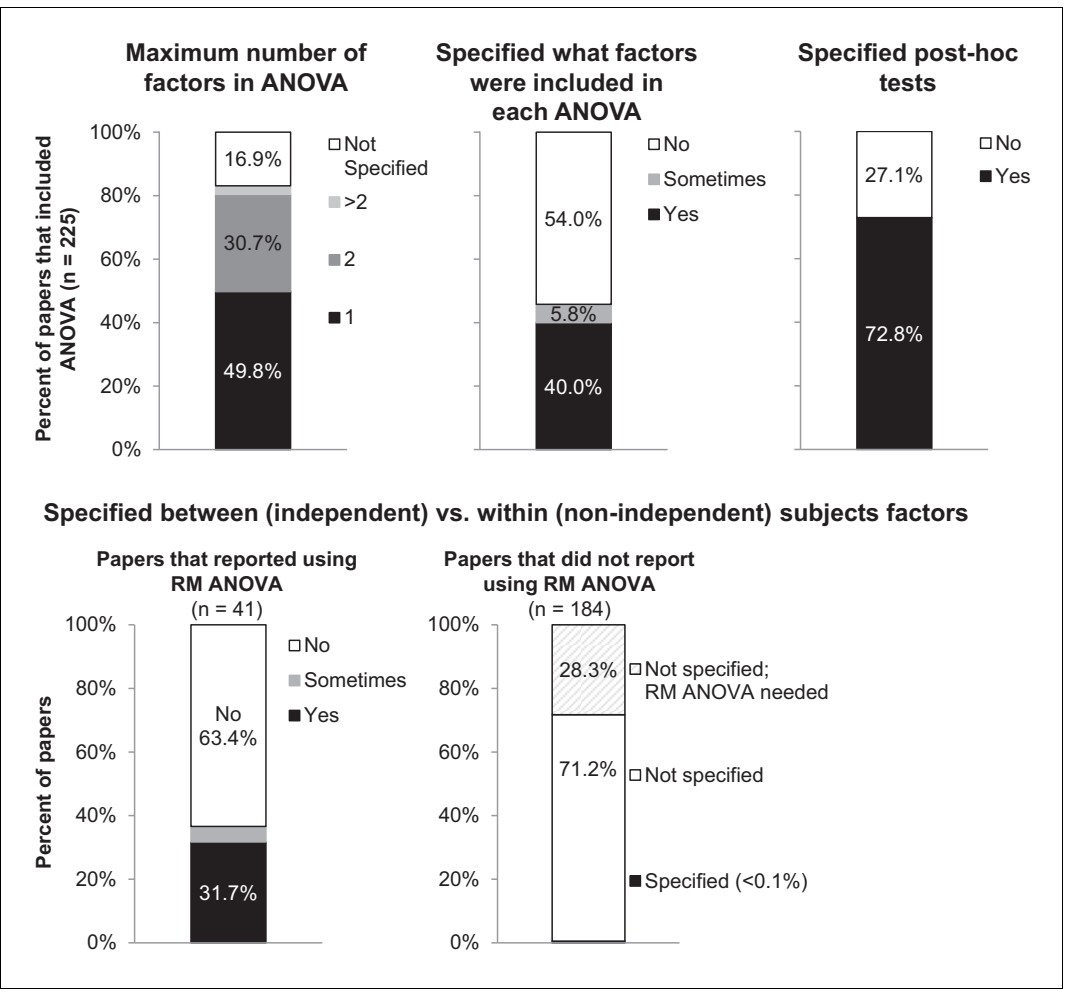

**Figure 2.** Many papers lack the information needed to determine what type of ANOVA was performed. The figure illustrates the proportion of papers in our sample that reported information needed to determine what type of ANOVA was performed, including the number of factors, the names of factors, and the type of post-hoc tests. The top panel presents the proportion of all papers that included ANOVA (n = 225). 'Sometimes' indicates that the information was reported for some ANOVAs but not others. The bottom row examines the proportion of papers that specified whether each factor was between vs. within-subjects. Papers are subdivided into those that reported using repeated measures ANOVA (n = 41), and those that did not report using repeated measures ANOVA (n = 184). RM: repeated measures.

DOI: https://doi.org/10.7554/eLife.36163.006

any information on paired vs. unpaired t-tests reported that the Student's t-test was used. While the term Student's t-test traditionally refers to the unpaired t-test for equal variances, we identified numerous instances where authors referred to a paired t-test as a Student's t-test. Due to this confusion, we did not assume that the Student's t-test was unpaired unless the authors included additional terms like unpaired t-test or independent samples t-test. *Figure 6* illustrates why clear reporting that allows readers to confirm that the correct test was used is essential. Unpaired and paired t-tests interpret

the data differently, test different hypotheses, use different information to calculate the test statistic and usually give different results. If the incorrect test is used, the analysis tests the wrong hypothesis and the results may be misleading. Without the original data, it is difficult to determine how the test results would have been different had the appropriate tests been used.For example, differencesbetween the results of the paired and unpaired t-tests depend on the strength of the correlation between paired data points (*Figure 7*), which is

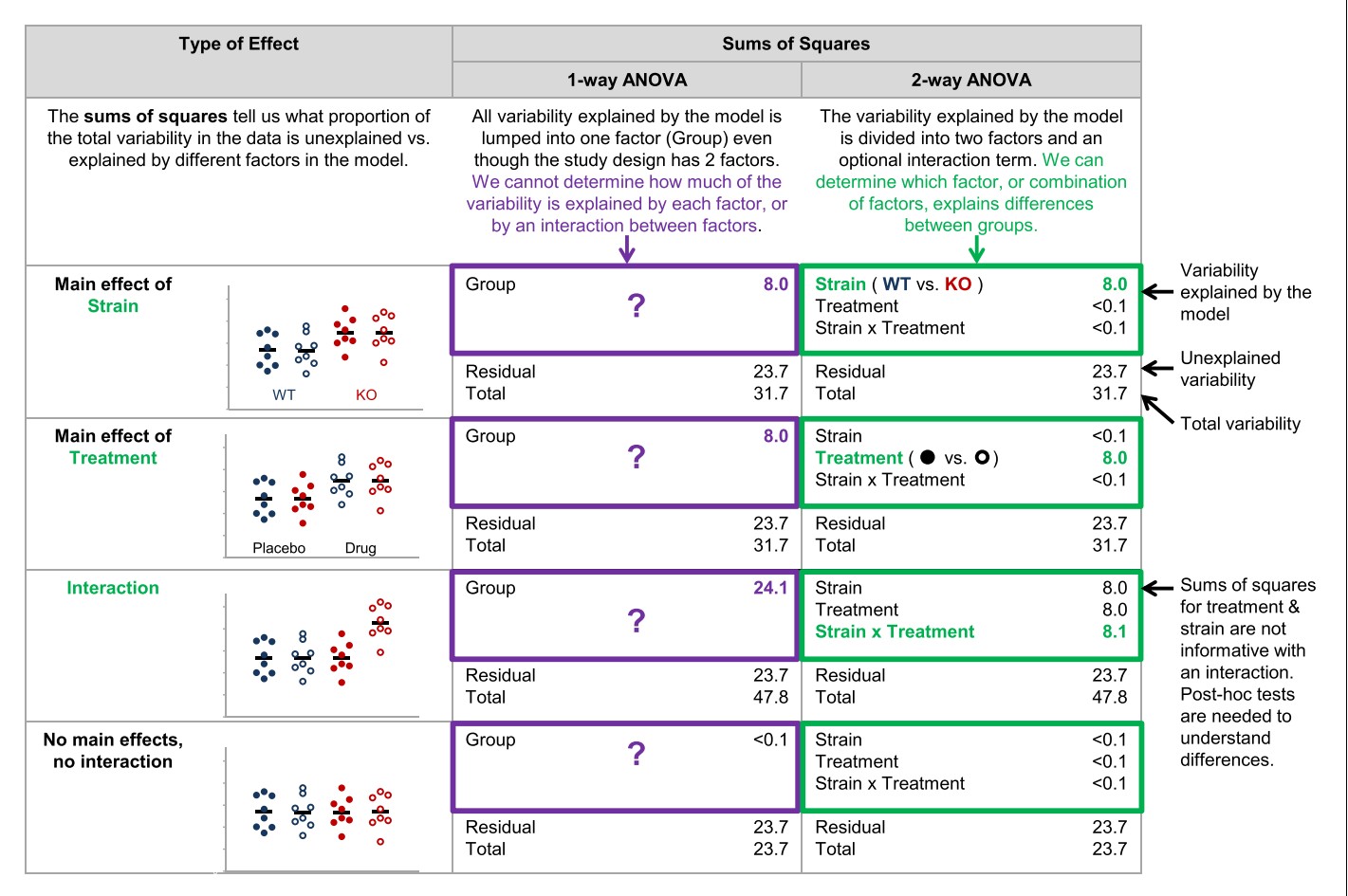

**Figure 3.** Why it matters whether investigators use a one-way vs two-way ANOVA for a study design with two factors. The two-way ANOVA allows investigators to determine how much of the variability explained by the model is attributed to the first factor, the second factor, and the interaction between the two factors. When a one-way ANOVA is used for a study with two factors, this information is missed because all variability explained by the model is assigned to a single factor. We cannot determine how much variability is explained by each of the two factors, or test for an interaction. The simulated dataset includes four groups – wild-type mice receiving placebo (closed blue circles), wild-type mice receiving an experimental drug (open blue circles), knockout mice receiving placebo (closed red circles) and knockout mice receiving an experimental drug (open red circles). The same dataset was used for all four examples, except that means for particular groups were shifted to show a main effect of strain, a main effect of treatment, and interaction between strain and treatment or no main effects and no interaction. One- and two-way (strain x treatment) ANOVAs were applied to illustrate differences between how these two tests interpret the variability explained by the model.
DOI: https://doi.org/10.7554/eLife.36163.007

difficult or impossible to determine from line graphs that only show summary statistics.

### Equal vs. unequal variance
When using unpaired t-tests, it is important to specify whether the test assumes that the variance is equal in both groups. The unpaired Student's t-test (also called the t-test for equal variances) assumes that the variance is similar in both groups, whereas the Welch's t-test (the t-test for unequal variances) assumes that variance differs between the two groups. If the variance is not estimated appropriately, the type I error is actually higher than advertised; this means that the null hypothesis is falsely rejected more often (*Ruxton, 2006*). Among the 155 papers that included unpaired t-tests, 64.5% (100/155) reported whether the test assumed equal variance for all unpaired t-tests and 6.5% (10/155) provided this information for some unpaired tests.

### Papers rarely contain information needed to verify the test result
Reporting the test statistic (ANOVA: F-statistic, t-tests: t-statistic), degrees of freedom and exact

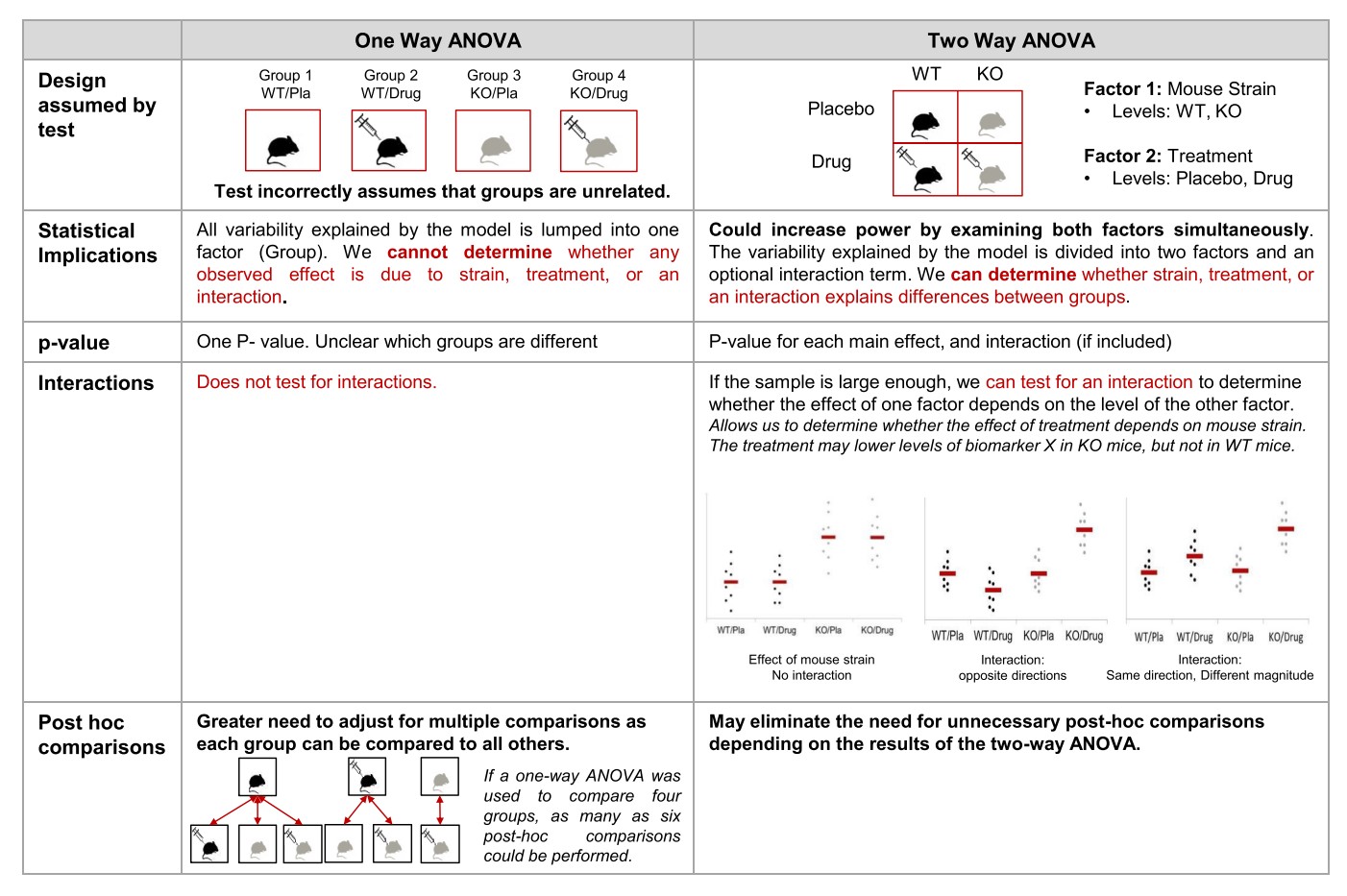

**Figure 4.** Additional implications of using a one-way vs two-way ANOVA. This figure compares key features of one- and two-way ANOVAs to illustrate potential problems with using a one-way ANOVA for a design with two or more factors. When used for a study with two factors, the one-way ANOVA incorrectly assumes that the groups are unrelated, generates a single p-value that does not provide information about which groups are different, and does not test for interactions. The two-way ANOVA correctly interprets the study design, which can increase power. The two-way ANOVA also allows for the generation of a set of p-values that provide more information about which groups may be different, can test for interactions, and may eliminate the need for unnecessary post-hoc comparisons. This figure uses an experimental design with four groups (wild-type mice receiving placebo, wild-type mice receiving an experimental drug, knockout mice receiving placebo and knockout mice receiving an experimental drug). See *Figure 2* for a detailed explanation of the material in the statistical implications section. KO: knockout; WT: wild-type; Pla: placebo.
DOI: https://doi.org/10.7554/eLife.36163.008

p-values are important for several reasons. First, these values allow authors, reviewers and editors to confirm that the correct test was used. This is particularly important when the statistical methods do not provide enough information to determine which type of ANOVA or t-test was performed, which was common in our dataset. Second, reporting the degrees of freedom allows readers to confirm that no participants, animals or samples were excluded without explanation. A recent study found that more than two thirds of papers using animal models to study stroke or cancer did not contain sufficient information to determine whether animals were excluded, while 7-8% of papers excluded animals without explanation (*Holman et al., 2016*). Biased exclusion of animals or observations can substantially increase the probability of false positive results in small sample size studies. Third, this information allows authors and readers to verify the test result. In an analysis of psychology papers, over half reported at least one p-value that did not match the test statistic and degrees of freedom: this error was sufficient to alter the study conclusions in one out of eight papers (*Nuijten et al., 2016*). Authors and journal editors can eliminate these errors by double checking their results, or using software

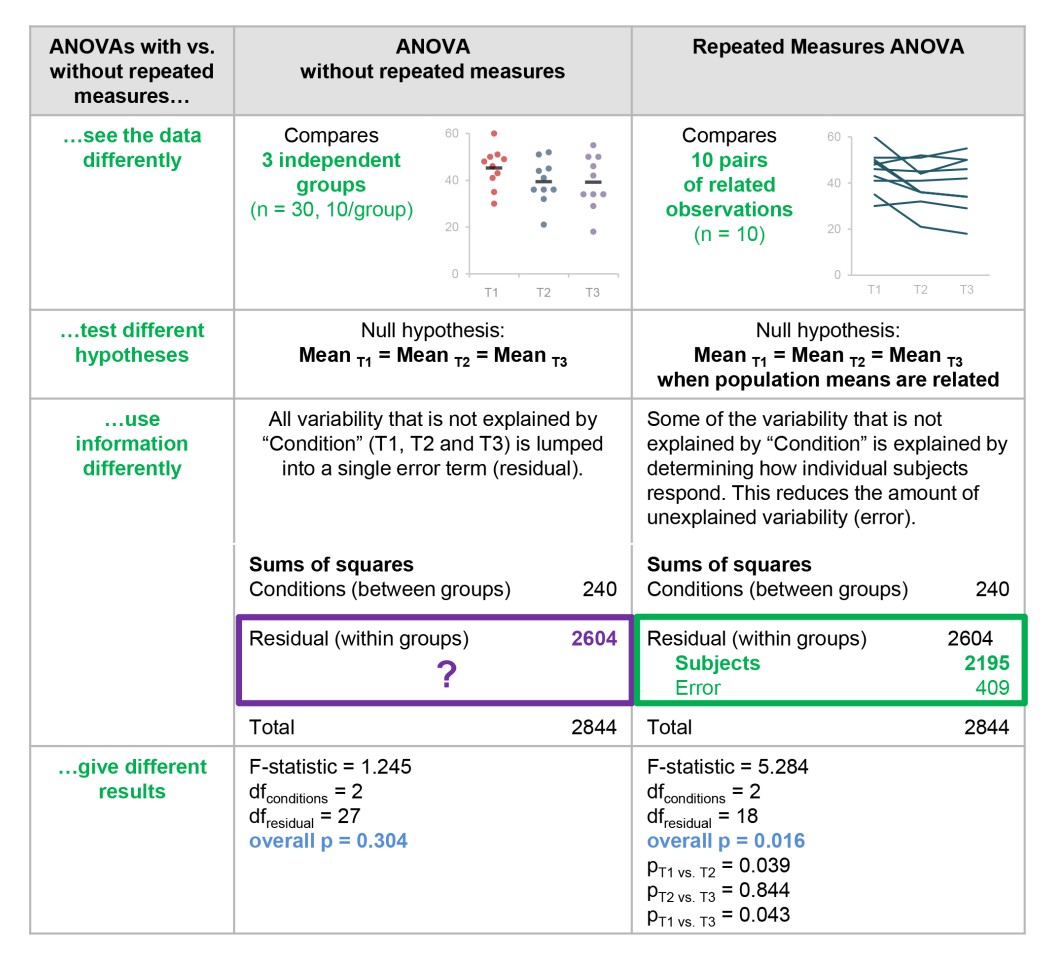

**Figure 5.** Why it matters whether investigators used an ANOVA with vs. without repeated measures. This figure highlights the differences between ANOVA with vs. without repeated measures and illustrates the problems with using an ANOVA without repeated measures when the study design includes longitudinal or non-independent measurements. These two tests interpret the data differently, test different hypotheses, use information differently when calculating the test statistic, and give different results.
DOI: https://doi.org/10.7554/eLife.36163.009

programs such as statcheck (*Eskamp and Nuijten, 2016*) to confirm that p-values match the reported test statistic and degrees of freedom. Unfortunately, the information needed to verify the test result is not reported in many papers.

Among the studies in our sample that included ANOVA (*Figure 8*), more than 95% failed to report the F-statistic or degrees of freedom. Moreover, 77.8% of these papers reported ranges of p-values (i.e., p>0.05, p<0.05, p<0.01), with just 50 papers reporting exact p-values (41 of 225 papers reported exact p-values for some ANOVAs, and 9 papers reported exact p-values for all ANOVAs). Among studies that included t-tests, 16 papers (8.9%) were excluded because abstractors were unable to determine what data were analyzed by

t-tests or to identify a two-group comparison. While most papers reported the sample size or degrees of freedom for some (16.6%) or all (76.7%) t-tests, t-statistics were missing from 95.7% of papers (*Table 1*). Moreover, 79.7% of the papers that included t-tests reported ranges of p-values instead of exact p-values.

## Moving towards a more transparent and reproducible future

This systematic review demonstrates that t-tests and ANOVA may not be so simple after all, as many papers do not contain sufficient information to determine why a particular test was selected, what type of test was used or to verify

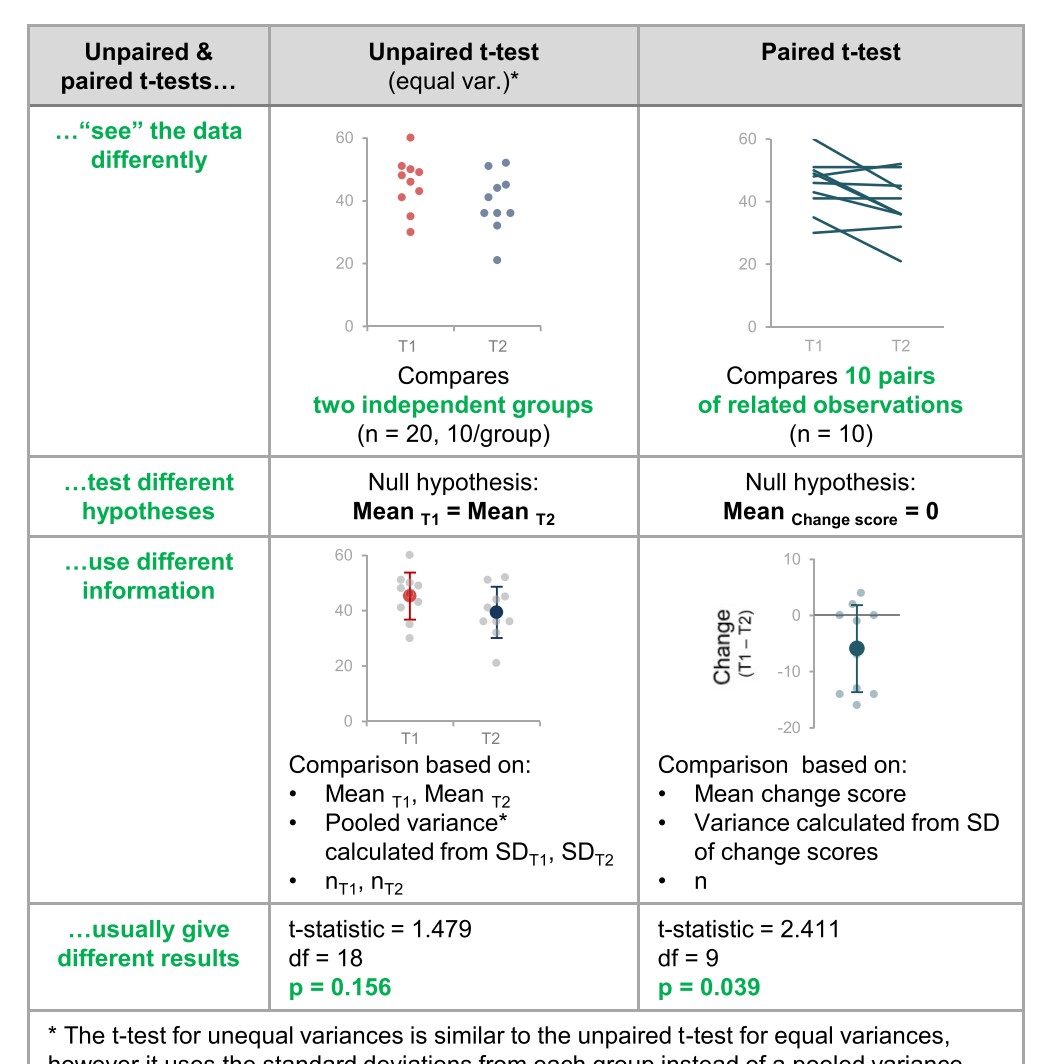

**Figure 6.** Why papers need to contain sufficient detail to confirm that the appropriate t-test was used. This figure highlights the differences between unpaired and paired t-tests by illustrating how these tests interpret the data differently, test different hypotheses, use information differently when calculating the test statistic, and give different results. If the wrong t-test is used, the result may be misleading because the test will make incorrect assumptions about the experimental design and may test the wrong hypothesis. Without the original data, it is very difficult to determine what the result should have been (see *Figure 6*).
DOI: https://doi.org/10.7554/eLife.36163.010

the test result. Selecting statistical tests that are not appropriate for the study design may be surprisingly common. Often, it is not possible to determine why statistical tests were selected, or whether other analyses may have provided more insight. Investigators frequently present data using bar or line graphs that only show summary statistics (*Weissgerber et al., 2015*) and raw data are rarely available. Many journals have recently introduced policies to encourage more informative graphics, such as dot plots, box

plots or violin plots, that show the data distribution (*Nature, 2017*; *Fosang and Colbran, 2015*; *Kidney International, 2017*; *PLOS Biology, 2016*; *Teare, 2016*) and may provide insight into whether other tests are needed. Non-parametric tests, such as the Mann-Whitney U test, Wilcoxon sign rank test and Kruskal Wallis test, may sometimes be preferable as sample sizes in basic biomedical science research are often too small to determine the data distribution. However, these tests are not commonly

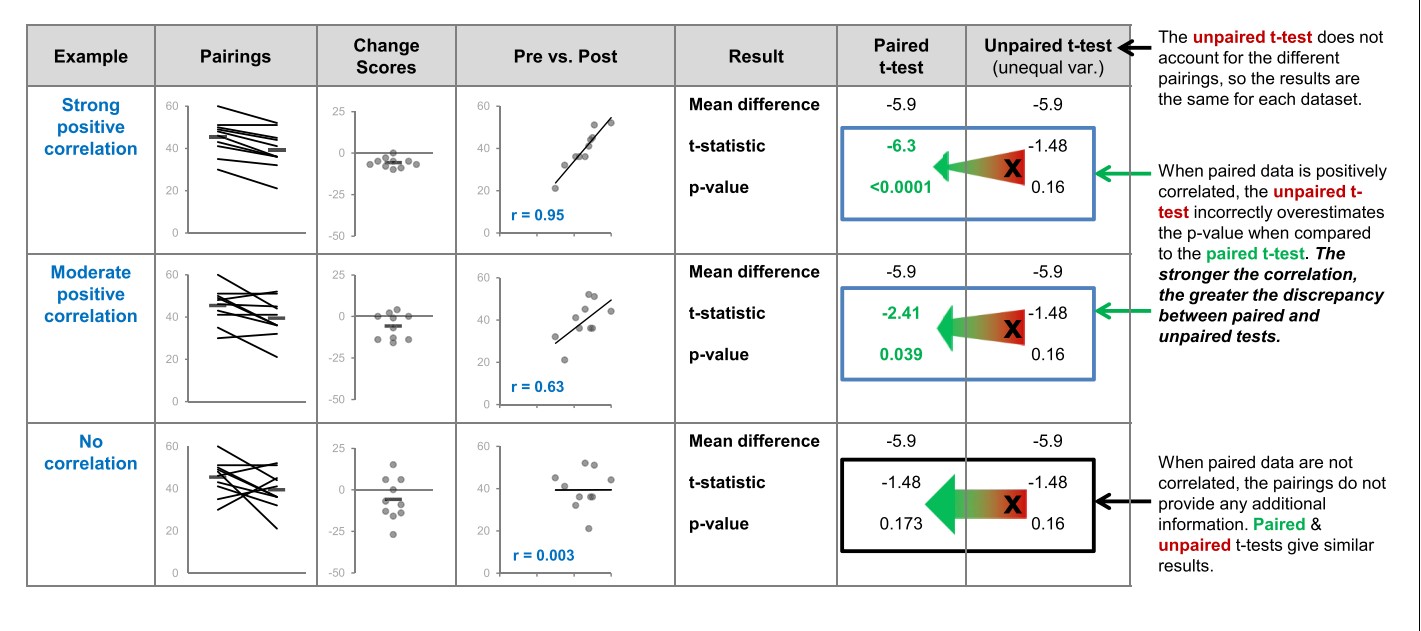

**Figure 7.** Differences between the results of statistical tests depend on the data. The three datasets use different pairings of the values shown in the dot plot on the left. The comments on the right side of the figure illustrate what happens when an unpaired t-test is inappropriately used to compare paired, or related, measurements. We expect paired data to be positively correlated – two paired observations are usually more similar than two unrelated observations. The strength of this correlation will vary. We expect observations from the same participant to be more similar (strongly correlated) than observations from pairs of participants matched for age and sex. Stronger correlations result in greater discrepancies between the results of the paired and unpaired t-tests. Very strong correlations between paired data are unusual but are presented here to illustrate this relationship. We do not expect paired data to be negatively correlated – if this happens it is important to review the experimental design and data to ensure that everything is correct.

DOI: https://doi.org/10.7554/eLife.36163.011

used. A systematic review of physiology studies showed that 3.8% used only non-parametric tests to compare continuous data, 13.6% used a combination of parametric and non-parametric tests, and 78.1% of studies used only parametric tests (*Weissgerber et al., 2015*). A recent primer *Hardin and Kloke, 2017* provides a brief overview of common statistical techniques, including non-parametric alternatives to t-tests and ANOVAs.

Recently, fields such as psychology have been moving away from a reliance on t-tests and ANOVAs towards more informative techniques, including effect sizes, confidence intervals and meta-analyses (*Cumming, 2014*). While p-values focus on whether the difference is statistically significant, effect sizes answer the question: "how big is the difference?" A published tutorial provides more details on how to calculate effect sizes for t-tests and ANOVA (*Lakens, 2013*). Presenting confidence intervals provides more information about the uncertainty of the estimated statistic (i.e., mean, effect size, correlation coefficient). A guide to robust statistical

methods in neuroscience examines a variety of newer methods to address the problems with hypothesis tests, and includes information on techniques that are suitable for studies with small sample sizes (*Wilcox and Rousselet, 2018*). Small, underpowered studies frequently produce inconclusive results that may appear to be contradictory. Meta-analyses combine these divergent results to provide a comprehensive assessment of the size and direction of the true effect (*Cumming, 2014*).

The findings of the present study highlight the need for investigators, journal editors and reviewers to work together to improve the quality of statistical reporting in submitted manuscripts. While journal policy changes are a common solution, the effectiveness of policy changes is unclear. In a study of life sciences articles published in *Nature* journals, the percentage of animal studies reporting the Landis 4 criteria (blinding, randomization, sample size calculation, exclusions) increased from 0% to 16.4% after new guidelines were released (*Macleod and The NPQIP Collaborative group,*

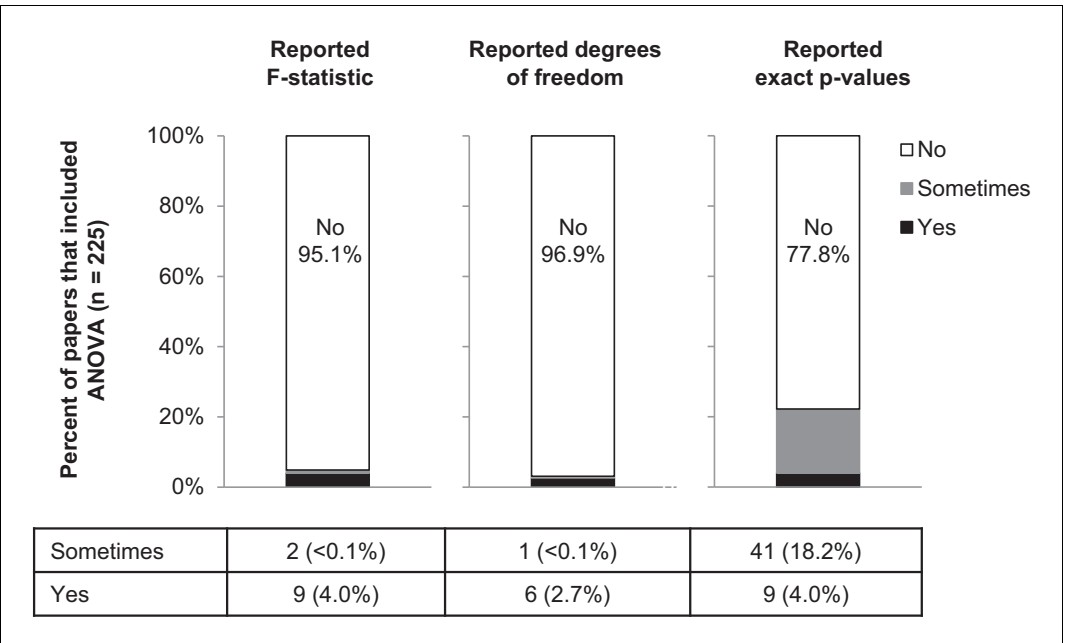

**Figure 8.** Few papers report the details needed to confirm that the result of the ANOVA was correct. This figure reports the proportion of papers with ANOVAs (n = 225) that reported the F-statistic, degrees of freedom and exact p-values. Sometimes indicates that the information was reported for some ANOVAs contained in the paper but not for others.
DOI: https://doi.org/10.7554/eLife.36163.012

*2017*). In contrast, a randomized controlled trial of animal studies submitted to PLoS One demonstrated that asking authors to complete the ARRIVE checklist at the time of submission had no effect (*Hair et al., 2018*). Some improvements in reporting of confidence intervals, sample size justification and inclusion and exclusion criteria were noted after *Psychological Science* introduced new policies, although this may have been partially due to widespread changes in the field (*Giofrè et al., 2017*). A joint editorial series published in the *Journal of Physiology* and

*British Journal of Pharmacology* did not improve the quality of data presentation or statistical reporting (*Diong et al., 2018*). Statistical reporting requirements are not standard practice for many fields and journals in basic biomedical science, thus limiting the ability of readers to critically evaluate published research.

Other possible solutions to improve reporting include strengthening and implementing reporting guidelines, and training editors, reviewers and investigators to recognize common problems. *Box 2* lists statistical information that

**Table 1.** Reporting of details needed to verify the results of a t-test.

|  | Reported t-statistic | Reported exact sample size or degrees of freedom | Reported exact p-values |
|---|---|---|---|
| No | 156 (95.7%) | 11 (6.7%) | 113 (69.3%) |
| Sometimes | 0 | 27 (16.6%) | 17 (10.4%) |
| Yes | 7 (4.3%) | 125 (76.7%) | 33 (20.2%) |

We analyzed the 179 papers in our sample that included t-tests to check if they reported the details that are needed to verify the results of these tests: we had to exclude 16 papers from this analysis because we were unable to determine what data were analyzed by t-tests or to identify a two-group comparison. Most of the papers (95.7%; 156/163) did not report the t-statistic (column 2) and over two-thirds (69.3%; 113/163) did not report exact p-values (column 4), but over three-quarters (76.7%; 125/163) reported the exact sample size or degree of freedom for all of the t-tests in the paper (column 3).
DOI: https://doi.org/10.7554/eLife.36163.013

should be reported when presenting data that were analyzed by t-tests and ANOVA. Some journals have already begun offering unlimited word counts for methods sections to improve reporting (*Nature, 2013*), while others suggest putting additional information in supplemental methods. Creating open-source software programs that calculate word counts for the results section after excluding statistical information may also facilitate transparent reporting of statistical results. Encouraging public data archiving or interactive graphics that include all data (*Ellis and Merdian, 2015*; *Weissgerber et al., 2016b*; *Weissgerber et al., 2017*) may also strengthen reporting and reproducibility while facilitating data re-use.

## Materials and methods

### Systematic review of literature

Methodology for the systematic review was similar to the approach outlined in our previous paper (*Weissgerber et al., 2015*). Physiologists perform a wide range of studies involving humans, animals and in vitro laboratory experiments; therefore we examined original research articles that were published in June 2017 in the top 25% of journals (n = 21) in the Physiology category in Journal Citation Reports as determined by 2016 journal impact factor; six journals that only publish review articles and one journal that did not publish a June issue were excluded. Each phase of screening and data abstraction was performed by two independent reviewers (TLW, OGV). Disagreements were resolved by consensus. Both reviewers screened all articles published in each journal between June 1 and June 30, 2017 to identify full length, original research articles (*Figure 1* and *Supplementary file 1*). Full text articles were then reviewed and papers were excluded if they did not include new data, did not have a continuous outcome variable, or did not include an analysis of variance (ANOVA) or t-test. Eligible manuscripts and supplements were reviewed in detail to evaluate the following questions according to a predefined protocol (*Supplementary file 2*). This systematic review was conducted in accordance with those elements of the PRISMA guideline (*Liberati et al., 2009*) that are relevant to literature surveys.

Questions asked for papers that included ANOVAs

1. What was the maximum number of factors included in any ANOVA performed in the paper?
2. Could the names of factors that were included in each ANOVA be determined from the text, tables or figures?
3. Did the paper report using a repeated measures ANOVA?
4. Did the authors specify whether each factor was included in the ANOVA as a between–subjects (independent) or within-subjects (non-independent) factor?
5. Did the paper specify which post-hoc tests were performed?
6. If ANOVA was performed but the paper did not mention including repeated measures, within-subjects factors, or non-independent factors, did the paper include any analyses that appeared to require a repeated measures ANOVA (i.e., longitudinal data or other non-independent data)?
7. If the paper reported using a maximum of one factor in ANOVAs, was ANOVA used to compare groups that could be divided into two or more factors?
8. Did the paper report F-statistics, degrees of freedom and exact p-values when describing ANOVA results?

Questions asked for papers that included t-tests

1. Did the paper specify whether paired or unpaired t-tests were used for each analysis?
2. Did the paper specify whether unpaired t-tests assumed equal or unequal variance for each analysis?
3. Did the paper report t-statistics, sample size or degrees of freedom, and exact p-value for data analyzed by t-test?

### Statistical analysis

Data are presented as n (%). The objective of this observational study was to assess standard practices for reporting statistical methods and results when data are analyzed by ANOVA; therefore no statistical comparisons were performed. Summary statistics were calculated using JMP (10.0.0, SAS Institute Inc., Cary, NC). Ethical approval was not required.

**Tracey L Weissgerber** is in the Division of Nephrology & Hypertension, Mayo Clinic, Rochester, Minnesota, United States, and QUEST – Quality | Ethics | Open

Science | Translation, Charité - Universitätsmedizin Berlin, Berlin Institutes of Health, Berlin, Germany
weissgerber.tracey@mayo.edu
http://orcid.org/0000-0002-7490-2600

**Oscar Garcia-Valencia** is in the Division of Nephrology & Hypertension, Mayo Clinic, Rochester, Minnesota, United States
https://orcid.org/0000-0003-0186-9448

**Vesna D Garovic** is in the Division of Nephrology & Hypertension, Mayo Clinic, Rochester, Minnesota, United States
http://orcid.org/0000-0001-6891-0578

**Natasa M Milic** is in the Division of Nephrology & Hypertension, Mayo Clinic, Rochester, Minnesota, United States and the Department of Medical Statistics & Informatics, Medical Faculty, University of Belgrade, Belgrade, Serbia

**Stacey J Winham** is in the Division of Biomedical Statistics & Informatics, Mayo Clinic, Rochester, Minnesota, United States
https://orcid.org/0000-0002-8492-9102

*Author contributions:* Tracey L Weissgerber, Conceptualization, Formal analysis, Supervision, Funding acquisition, Validation, Investigation, Visualization, Methodology, Writing—original draft, Project administration, Writing—review and editing; Oscar Garcia-Valencia, Investigation, Visualization, Methodology, Writing—review and editing; Vesna D Garovic, Supervision, Funding acquisition, Writing—review and editing; Natasa M Milic, Conceptualization, Validation, Methodology, Writing—review and editing; Stacey J Winham, Conceptualization, Funding acquisition, Methodology, Writing—review and editing

*Competing interests:* The authors declare that no competing interests exist.

Funding

| Funder | Grant reference number | Author |
| --- | --- | --- |
| American Heart Association | 16GRNT30950002 | Tracey L Weissgerber |
| National Center for Advancing Translational Sciences | UL1 TR000135 | Tracey L Weissgerber |
| Mayo Clinic | Robert W Fulk Career Development Award | Tracey L Weissgerber |
| National Cancer Institute | R03-CA212127 | Stacey J Winham |
| | Walter and Evelyn Simmers Career Development Award for Ovarian Cancer Research | Stacey J Winham |

The funders had no role in study design, data collection and interpretation, or the decision to submit the work for publication.

**Decision letter and Author response**
Decision letter https://doi.org/10.7554/eLife.36163.020
Author response https://doi.org/10.7554/eLife.36163.021

## Additional files

### Supplementary files
• Supplementary file 1. Number of articles examined by journal.
DOI: https://doi.org/10.7554/eLife.36163.014

• Supplementary file 2. Abstraction protocol for systematic review.
DOI: https://doi.org/10.7554/eLife.36163.015

• Transparent reporting form
DOI: https://doi.org/10.7554/eLife.36163.016

• Reporting standard 1 PRISMA 2009 checklist.
DOI: https://doi.org/10.7554/eLife.36163.017

### Data availability
All data from the systematic review has been uploaded with the manuscript, along with the abstraction protocol.

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
