## [Decision Letter]

Thank you for submitting your article 'Why We Need to Report More Than "Data were Analyzed by t-tests or ANOVA"' to *eLife* for consideration as a Feature Article. Your article has been reviewed by three peer reviewers, including Dawn Teare as the Reviewing Editor and Reviewer #1, and the evaluation has been overseen by Peter Rodgers, the *eLife* Features Editor. The following individuals involved in review of your submission have agreed to reveal their identity: Andrew Althouse (Reviewer #2); Glenn Begley (Reviewer #3).

The reviewers have discussed the reviews with one another and the Reviewing Editor and Features Editor have drafted this decision to help you prepare a revised submission.

Summary:

This paper is a systematic review of 328 papers published in June 2017 in 21 physiology journals. The authors focused on the use of t-tests and ANOVA. They conclude that key information was routinely absent. Similar studies have been published in the past that describe the problems with statistical analysis of scientific papers. However, with the change in Guidelines to Authors instituted by many journals over recent years, it might have been hoped that the situation had improved. Sadly, that does not appear to be the case. The paper includes several figures that demonstrate the challenges that arise when data is inappropriately analysed: these are very useful. The authors are to be congratulated on a clear, well-written, compelling study.

Essential revisions:

Reviewer #1:

a) Given the authors are stating that reporting initiatives are urgently needed, it is a shame that they did not use the PRISMA guidelines to draft the manuscript. There are many reporting initiatives out there, we don't need new ones but rather we need to know how the existing ones can be improved. This paper has been written without much reference to those other initiatives.

b) The manuscript selects instances where a t-test or an ANOVA have been performed. There is no mention of the use of non-parametric tests. Although there is some discussion of how to optimally use an ANOVA to examine interactions, the work tends to focus on being able to replicate exactly what has been done by providing more details. The article would benefit from some discussion of how to select the optimal analysis strategy given a research hypothesis.

c) The checklist provided in box 2 does not have prompts for effect sizes and confidence intervals to be reported. Many of the new reporting guidelines that have been proposed recently directly encourage these to be reported rather than the F-test values and degrees of freedom. This is also missing from the ANOVA section. Again the article would benefit from some discussion of comparative statistics and confidence intervals.

d) The final section ('Transparency: Where should we draw the line?') is missing the idea of reproducibility. Transparency is important for reproducibility. I would suggest including the following recommendation: There should be sufficient detail in a paper so that: i) a reader can understand why the author has selected the analysis methods they have; ii) a reader can repeat the analysis and get the same answer (if they have the raw data).

e) Please expand the figure captions to better explain what is shown in each figure.

---

## [Author Response]

Essential revisions:Reviewer #1:a) Given the authors are stating that reporting initiatives are urgently needed, it is a shame that they did not use the PRISMA guidelines to draft the manuscript. There are many reporting initiatives out there, we don't need new ones but rather we need to know how the existing ones can be improved. This paper has been written without much reference to those other initiatives.

We have added a statement to the Materials and methods that the systematic review was conducted in accordance with all relevant aspects of the PRISMA guidelines and included a PRISMA checklist. We have also integrated the SAMPL guidelines for statistical reporting in the Introduction. The SAMPL guidelines are very general and may not include adequate information for scientists with little or no statistical training to determine what details should be reported. This paper is designed to provide basic biomedical scientists with more information about why it is important to include additional information so that readers can determine whether an appropriate type of ANOVA or t-test was performed. While we focus on t-tests and ANOVA because they are the most common, the same principles apply to other types of tests.

We always follow all relevant aspects of the PRISMA guidelines as part of our standard protocol for conducting systematic reviews. However, it is important to note that large portions of the guidelines do not apply to literature surveys or to systematic reviews without a meta-analysis. For example, search terms are not included because no selective electronic search was performed. Journals were identified by selecting the top 25% of journals, according to 2016 impact factor, in the physiology category of the Journal Citation Reports database. Articles were then identified by examining journal websites to identify all articles in issues published in June 2017. This process was completed by two independent reviewers. The PRISMA guidelines recommend a structured Abstract, however this contradicts this journal’s requirements for features articles. Elements such as PICO, risk of bias assessments, and reporting of elements relevant to meta-analyses are not relevant to this literature survey.

b) The manuscript selects instances where a t-test or an ANOVA have been performed. There is no mention of the use of non-parametric tests. Although there is some discussion of how to optimally use an ANOVA to examine interactions, the work tends to focus on being able to replicate exactly what has been done by providing more details. The article would benefit from some discussion of how to select the optimal analysis strategy given a research hypothesis.

We chose to focus on t-tests and ANOVA for two reasons. First, while t-tests and ANOVA are very common, our previous work has shown that non-parametric tests are not. We examined articles published in the top 25% of physiology journals during a three-month period in 2014 (Weissgerber et al., 2015). We found that 78.1% of studies used only parametric tests to compare continuous data (i.e. t-tests and ANOVA), 13.6% used a combination of parametric and non-parametric tests, and 3.8% used only non-parametric tests. We have now described this data in the manuscript. Second, there are many different types of ANOVAs and t-tests; therefore additional detail is needed to determine which test was performed. In contrast, the name of the non-parametric test (i.e. Kruskal-Wallis, Mann Whitney, Wilcoxon Rank Sum, Wilcoxon Signed Rank) is sufficient because these tests do not have variations.

We understand the reviewers’ desire for a single paper that outlines the problems with the reporting of existing analyses, addresses the misconceptions that lead to these reporting errors and shows how to correct them, and also describes how to select more informative or optimal tests. However, it is important to be realistic about what can be accomplished in a single paper designed for basic biomedical scientists, many of whom have limited or no statistical training. There are several papers that address the problems with t-tests and ANOVA and describe how to select more informative alternatives. Although these papers tend to be quite long, the authors frequently note that they are intended to supplement existing statistical knowledge and are not a substitute for statistical training. We have rewritten the last section of the paper to briefly address other approaches and referred readers to existing resources for additional information. A shift away from hypothesis testing in basic biomedical science will require retraining the scientific workforce, which will take time. It is likely that most investigators will continue to perform t-tests and ANOVA in the near future; hence, efforts to improve reporting for these techniques are needed.

c) The checklist provided in box 2 does not have prompts for effect sizes and confidence intervals to be reported. Many of the new reporting guidelines that have been proposed recently directly encourage these to be reported rather than the F-test values and degrees of freedom. This is also missing from the ANOVA section. Again the article would benefit from some discussion of comparative statistics and confidence intervals.

We have updated Box 2 to include effect sizes and confidence intervals and added a section at the end of the manuscript that addresses other statistical approaches and provides links to papers that examine these resources in detail. This includes a citation on how to calculate effect sizes for t-tests and ANOVA.

d) The final section ('Transparency: Where should we draw the line?') is missing the idea of reproducibility. Transparency is important for reproducibility. I would suggest including the following recommendation: There should be sufficient detail in a paper so that: i) a reader can understand why the author has selected the analysis methods they have; ii) a reader can repeat the analysis and get the same answer (if they have the raw data).

These are both included in the SAMPL guidelines, which we have now mentioned in the Introduction. Additionally, these points are also both mentioned in the first sentence of the final section of the paper, which has been extensively revised and is now entitled “Moving towards a more transparent and reproducible future”. This section also includes a brief discussion on more informative analysis techniques and links to resources and integrates the concept of reproducibility.

e) Please expand the figure captions to better explain what is shown in each figure.

We have added additional descriptors to figure legends.